# Intra-urban differentials in the exclusive use of hygienic methods during menstruation among young women in India

**Aditya Singh**[1,2]*, **Mahashweta Chakrabarty**[1]*, **Rakesh Chandra**[3], **Sourav Chowdhury**[4], **Shivani Singh**[5]

1 Banaras Hindu University, Varanasi, India, 2 Girl Innovation, Research, and Learning Centre, Population Council, New York, NY, United States of America, 3 Tata Institute of Social Sciences, Mumbai, India, 4 Raiganj University, Raiganj, India, 5 India Health Action Trust, Lucknow, India

* adityasingh@bhu.ac.in (AS); mahashweta.c1997@gmail.com (MC)

**Data Availability Statement:** The study utilizes secondary sources of data that are freely available in the public domain through https://dhsprogram.

## Abstract

Menstrual hygiene among women is a critical public health issue in urban India, but it remains understudied and under-researched. However, to our knowledge, no national level study in India has yet examined the differentials in the exclusive use of hygienic methods among young women (aged 15–24) in urban India. This study attempts to fill this gap by analysing biodemographic, socioeconomic, and geographic differentials in the exclusive use of hygienic methods among these women. We analysed data on 54561 urban women aged 15–24 from National Family Health Survey-5, 2019–21. We used binary logistic regression to examine differentials in the exclusive use of hygienic methods. To examine spatial variation, we mapped exclusive use of hygienic methods across Indian states and districts. The study found that two-thirds of young women in urban India reported exclusive use of hygienic methods. However, there was significant geographic heterogeneity observed at both state and district levels. In states such as Mizoram and Tamil Nadu, the use of hygienic methods was over 90%, while in Uttar Pradesh, Bihar, Chhattisgarh, and Manipur, it was less than 50%. The district-level variation in exclusive use of hygienic methods was even more striking. In many states, districts with extremely low exclusive use (less than 30%) were located in close proximity to districts with high exclusive use. Being poor, uneducated, Muslim, having no mass media exposure, living in the north and central regions, not having a mobile phone, getting married before 18, and having an early experience of menarche were associated with lower exclusive use of hygienic methods. In conclusion, substantial biodemographic, socioeconomic, and geographic differentials in the exclusive use of hygienic methods suggest the need for context-specific behavioural interventions. Mass media campaigns and targeted distribution of subsidized hygienic methods could help reduce the existing inequities in the exclusive use of hygienic methods.

com/methodology/survey/survey-display-541.cfm.
Those who wish to access the data may register at
the above link and thereafter can download the
required data free of cost.

**Funding:** The authors received no specific funding
for this work.

**Competing interests:** The authors have declared
that no competing interests exist.

## Introduction

Developing countries, including India, are witnessing rapid growth in their urban population. India's urban population has increased from 17.0% in 1951 to 34.9% in 2020 (285 million), with an annual growth of 2.3% between 2015 and 2020 [1]. Although cities are engines of economic growth and bring significant improvements in urban residents' health and living standards, they are also much more unequal than rural areas [2, 3]. Evidence suggests that an urban advantage in health and healthcare use exists when comparing averages of urban and rural areas as rural residents had lower health information access and use [4–7]. However, significant intra-urban heterogeneity is also revealed when examining these indicators across urban areas of a country [8, 9]. It has been evidenced that for some specific health outcomes and health services utilization indicators, certain urban population groups fare even more poorly than the rural population [10, 11]. However, government policies and programs often pay less attention to these intra-urban inequalities in health and healthcare use. Instead, their focus primarily remains on improving the health outcomes of rural communities. It might be owing to policymakers' strong belief in the so-called 'urban advantage' theory, which holds that urban areas have an advantage over rural areas in terms of health and wellbeing [7]. Thus, the issue of intra-urban differentials in health outcomes and healthcare use gets relegated to an issue of secondary concern.

Menstruation, a natural biological event signifying the start of a woman's reproductive period, affects billions of girls and women worldwide [12, 13]. It is crucial to the dignity and wellbeing of women and girls to maintain adequate cleanliness during the period of menstruation in order to limit the risk of contracting reproductive tract infections [14, 15]. While some women and girls use tampons, sanitary pads, and menstrual cups, others use cloths, paper, ash, plant leaves, and many other materials to absorb menstrual blood and maintain personal cleanliness during menstruation [16, 17]. The first three methods of menstrual hygiene management (tampons, sanitary pads, menstrual cups) are commonly classified as 'hygienic' in the previous studies and reports [18, 19]. Although the established classification scheme is suitable for analytical purposes, it has the potential limitation of not being able to capture the full range of women's experiences and perspectives regarding menstrual hygiene. It must be emphasized here that the use of this classification in this paper is solely intended for analytical purposes and is not intended to stigmatize or marginalize women who use alternative menstrual hygiene products/methods. The authors are committed to using language that is precise, inclusive, and respectful of the diversity of women's experiences with menstrual hygiene.

A growing body of evidence from low- and middle-income countries suggests that millions of women and girls resort to manage their periods with proper hygiene and dignity [20, 21]. This puts such women at a greater risk of reproductive tract infections, urinary tract infections, and other related problems [15, 22, 23]. Moreover, menstruation and menstrual hygiene-related insecurity contribute to school absenteeism and dropout risk for millions of girls and women [24–26]. The links between unmet menstrual health and hygiene requirements and RTIs and school absence are complex and mixed though [20, 27, 28]. Also, the shame and stigma attached to menstruation result in restrictions, prohibitions, and exclusion from public life, stopping women reach their full potential [29, 30]. It is therefore important that women are able to manage their menstruation with proper hygiene and dignity.

There is a significant body of literature documenting urban-rural dichotomy in the utilization of hygienic methods in developing countries, but little is known about the emerging heterogeneity in the use of hygienic methods within urban areas [15, 31–34]. In the Indian context, several small-scale studies based on individual cities have documented the utilization of hygienic methods during menstruation [35, 36]. A considerable amount of literature has

been published to understand the knowledge, perceptions, and practices regarding menstrual hygiene, focusing on young women in India. However, they are mainly micro-level community-based studies limited to small geographical areas [31, 35–39]. Previous national-level studies in India have consistently shown that level of education, household wealth, mass media exposure, and place of residence are important determinants of the use of hygienic methods during menstruation [32, 33, 40, 41]. However, none has examined the factors affecting the exclusive use of hygienic methods among young urban women at the national level.

Several previous studies have reported that adolescent and young women in South Asian countries like India may find menstruation particularly challenging because of pervasive stigma [17, 42–46]. Most people consider menstruating women to be dirty or filthy [47, 48]. Having a menstrual period is stigmatising and restricts a woman's ability to go about her daily life and keep herself clean [49–51]. Due to lack of autonomy, and disposable income menstruating adolescent girls and young women face evident risks to their health, safety, and quality of life when such limitations are imposed [20, 52]. The difficulties of period management in India can only be overcome if more is known about the menstrual hygiene practices of young women in the nation [44].

The urban population in Indian cities is socioeconomically and demographically quite diverse. Therefore, a crucial prerequisite for improving the utilization of safe and hygienic menstrual methods among young women in urban India is to ascertain the specific groups of women and girls who are at a disadvantage. Using the latest data available from the National Family Health Survey series, this study examines the biodemographic, socioeconomic and geographic differentials in the exclusive use of hygienic methods among women aged 15–24 years residing in urban India.

## Data and methods

This study used data from the National Family Health Survey (2019–2021). The NFHS is a nationally representative cross-sectional survey that collects data on various health-related issues, including maternal and child care, domestic violence, reproductive health and family planning [19]. The details of the sampling procedure and sample size are available in the national and state reports [19]. NFHS-5 interviewed 724,115 women of the reproductive age group (15–49 years) from 636,699 households in 28 states and 8 Union territories (UTs). In this paper, 54,561 urban women aged 15–24 from 28 states and 8 UTs were included in the analysis (see Fig 1 for the sample selection process).

### Conceptual framework

This study examines urban Indian women's use of hygienic methods and influencing factors. A framework was developed using evidence from the existing literature on factors associated with using hygienic methods during menstruation [33, 41, 53–56] (see Fig 2). The framework showed pathways through which various factors might affect the exclusive use of hygienic methods among urban Indian women. It was hypothesized that the exclusive use of hygienic methods was affected by various factors: demographic factors, socioeconomic factors, geographic factors, and factors related to exposure to information and services. The list of variables considered for analysis is provided later in this section.

### Variables

**Dependent variable.** The NFHS-5 included a question with multiple responses to determine the methods of protection used by respondents to prevent bloodstains during their menstrual period. The question comprised seven response categories, including menstrual cups,

Total women interviewed between the age group 15-49 years (in NFHS-5, 2019-21, India), n=724,115

544,580 women excluded because they belong to rural areas of India.

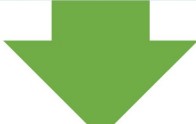

Total women (15-49 years) from urban India in NFHS-5, n=179,535

124,974 urabn women aged 25-49 years excluded because menstrual hygiene questions were not asked to them.

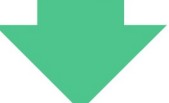

Urban women aged 15-24 years who were asked menstrual hygiene questions, n=54,561

**Fig 1. Flow chart showing the process of selection of urban women (15–24 years) sample for the current study.**

locally made napkins, sanitary napkins, tampons, cloth, nothing, and others. For the purpose of analysis, these responses were grouped into two categories: a) menstrual cups, locally made napkins, sanitary napkins, and tampons, and b) cloth, nothing, and others. Based on these categories, a binary outcome variable was created with women who used materials included in category 'a' only considered as "exclusive users of hygienic methods" (coded as '1'). Women who used materials included in category 'b' or a combination of materials from both categories were grouped as "non-exclusive users of hygienic methods" (coded as '0').

**Independent variables.** Several relevant socioeconomic and demographic predictors (including "respondent's current age", "age at menarche", "age at marriage", "women's education", "respondents' social group", "religion", "household wealth status", "region of residence", "types of home", "exposure to mass media", "discussion on menstrual hygiene with healthcare workers", "respondent's working status", "ownership of a bank account", and "mobile phone" were included in the analysis. The selection of variables was based on existing research on menstrual hygiene management and availability variables in the NFHS-5 dataset [11, 33, 32, 41, 53]. Table 1 describes the explanatory variables used in this study, their definitions and categories.

## Statistical analysis

The analysis starts with examining background characteristics of young women (aged 15–24 years) in urban India sampled in the NFHS-5. Subsequently, bivariate analysis was performed

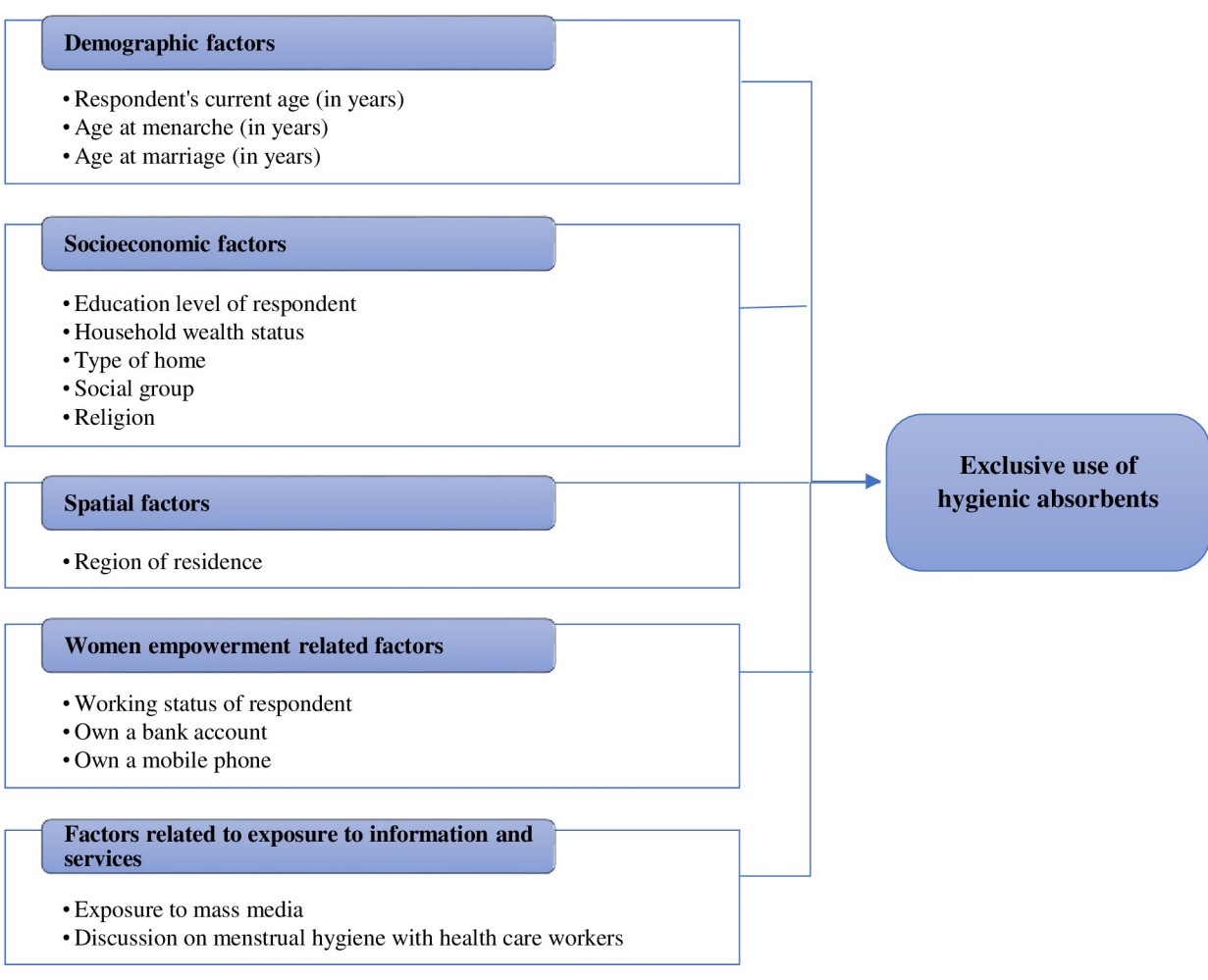

**Fig 2. Conceptual framework: Factors affecting exclusive use of hygienic methods.**

to scrutinize the disparity in the dependent variable across the respondents' demographic attributes and geographical factors. Thereafter, the multivariable binary logistic regression model was utilized to identify the correlates of the exclusive utilization of hygienic methods. This particular model was deemed appropriate for analysis since the response variable was dichotomous in nature [57]. Prior to conducting the multivariable binary logistic regression analysis, we first performed a Chi-squared test to evaluate the association between the outcome and each predictor variable. This initial step allowed us to eliminate any independent variables that did not demonstrate a statistically significant bivariate relationship with the outcome. All predictors that yielded a statistically significant result ($p<0.05$) in the Chi-squared testing were kept in the multivariable binary logistic regression. Given that our study involved several explanatory variables that could potentially exhibit correlation with one another, we also conducted an assessment of multicollinearity using variance inflation factors (VIF). The results of the logistic regression analysis were presented as odds ratios, accompanied by their corresponding p-values and 95% confidence intervals. All statistical analyses and modelling procedures were performed using Stata16 software [58]. In addition, ArcMap 10.5 software was used to create maps depicting the geographical distribution of the outcome variable at the state and district levels [59].

**Table 1. Operational definition of variables used in the study.**

| Variables | Description |
|---|---|
| **Age at menarche (in years)** | Age at menarche is the age when a woman has her first menstrual period. For the purpose of this study, we divided it into four categories: "less than 12 years" (coded as 1), "13–15 years" (coded as 2), "more than 16 years" (coded as 3), and for those who could not remember their age at menarche, it was coded as "do not know" (coded as 0). |
| **Age at marriage (in years)** | Age at marriage was classified into three categories- 'marriage before 18 years' (coded as 0), 'marriage after 18 years' (coded as 1), and 'not married' (coded as 2) |
| **Education level of respondent** | The education level variable in this study represents the highest level of education attained by respondents, categorized into four groups: 'no education' (coded as 0), 'primary' (coded as 1), 'secondary' (coded as 2), and 'higher' (coded as 3). |
| **Household wealth status** | The wealth index measures the socioeconomic status of a household. It is a composite index of household amenities and assets. In NFHS-5, every household is given a score based on the number of consumer goods they own. A total of 33 assets and housing characteristics were taken into consideration to prepare a factor score using Principal Component Analysis. After that, this factor score is divided into five equal categories: 'poorest' (coded as 1), 'poorer' (coded as 2), 'middle' (coded as 3), 'richer' (coded as 4), and 'richest' (coded as 5), each with 20% of the population. |
| **Social groups** | The entire population of our study sample is divided into four social groups: 'Scheduled Caste' (coded as 1), 'Scheduled Tribe' (coded as 2), 'Other Backward Classes' (coded as 3), 'Others' (often referred to as General) (coded as 4). |
| **Religion** | For the purpose of the study, we have recoded religions into four categories–'Hindu' (coded as 1), 'Muslim' (coded as 2), 'Christian' (coded as 3), and 'Others' (coded as 4). Others include all religious groups other than Hindu, Muslim, and Christian. |
| **Region of residence** | To construct this variable, Indian states and UTs are grouped into six categories. 'Northern' (coded as 1) includes Jammu & Kashmir, Ladakh, Himachal Pradesh, Punjab, Rajasthan, Haryana, Uttarakhand, Chandigarh (Union Territory—UT) and Delhi; 'central' (coded as 2) includes the states of Uttar Pradesh, Madhya Pradesh and Chhattisgarh; 'eastern' (coded as 3) includes the states of Bihar, Jharkhand, West Bengal and Odisha; 'western' (coded as 4) includes the states of Gujarat, Maharashtra, Goa and UTs of Dadra & Nagar Haveli and Daman & Diu; 'southern' (coded as 5) includes the states of Kerala, Karnataka, Andhra Pradesh, Tamil Nadu and the UTs of Andaman & Nicobar Islands, Pondicherry and Lakshadweep); 'northeastern' (coded as 6) includes the states of Sikkim, Assam, Meghalaya, Manipur, Mizoram, Nagaland, Tripura, and Arunachal Pradesh. This classification has been used by many previous studies [19] |
| **Type of home** | Based on the woman's relationship with the household head, the type of home in which she resided was categorized as a 'marital home' (coded as 1) (wife, daughter-in-law, or sister-in-law of the household head), 'natal home' (coded as 2) (daughter, granddaughter, or niece of the household head), or 'other's home' (coded as 3) (e.g., non-relatives such as domestic servants working in the household, orphans, deserted young women). |
| **Working status of the respondent** | Working status indicates the employment condition of the respondent. A dichotomous variable is formed: 'not working' (coded as 1) and 'working' (coded as 2) |

*(Continued)*

**Table 1.** (Continued)

| Variables | Description |
|---|---|
| **Exposure to mass media** | Three questions were asked to women in the NFHS-5 survey. They are i) how often they read newspapers/magazines, ii) how often they watch television, and iii) how often they listen to the radio. The responses are 'almost every day', 'at least once a week', 'less than once a week', and 'not at all'. Based on these responses, a composite index is computed, and divided into four categories: 'no exposure' (coded as 0) if the respondent is not exposed to any mass media; 'low exposure' (coded as 1) if a respondent is exposed to any one type of mass media; 'medium exposure' (coded as 2) if the respondent is exposed to any two types of mass media; 'high exposure' (coded as 3) if the respondent is exposed to all three types of mass media. |
| **Discussed menstrual hygiene with healthcare workers (in last three months)** | Questions were asked to the respondents in NFHS-5 are- i) in the last three months, if the respondent has met with any health worker-including an auxiliary nurse midwife (ANM), accredited social health activist (ASHA), *Anganwadi* worker (AWW), also known as Integrated Child Development Services worker, multipurpose worker (MPW), or any other community health worker; and ii) if they have discussed menstrual hygiene during the meeting. If the respondent did not discuss menstrual hygiene with healthcare workers, they are coded as 0; if discussed, 1. |
| **Own a bank account** | Whether a respondent owns a bank/savings account by herself–'yes' (coded as 1); 'no' (coded as 0) |
| **Own a mobile phone** | Whether a respondent owns a mobile phone by herself–'yes' (coded as 1); 'no' (coded as 0) |

# Results

## Respondent characteristics

This study sample includes 54,561 urban 15-24-year-old women. Two-thirds of the sampled women were unmarried (see Table 2). Most women experienced menarche between 13–15. Seventy percent of females identified as Hindu. Females from the Other Backward Classes made up almost half of the sampled population (OBCs). Around 8% of women had no exposure to mass media. About 45% of the respondents lived in India's north and central regions.

## Prevalence of hygienic methods use by background characteristics

Table 3 presents the proportion of urban women exclusively using hygienic methods by their background characteristics. The exclusive use of hygienic methods is comparatively higher among urban women who experienced menarche after the age of 16, than among those who experienced menarche earlier. The use of exclusive hygienic methods was relatively higher among those who were either unmarried (71.0%) or married after the legal age of 18 years (63.9%) than those who were married before the legal age (52.3%).

The exclusive use of hygienic methods among those having higher education was several times higher (about 79.0%) than those who never went to school (33.4%). It is observed that the exclusive use of hygienic methods was considerably higher among Christians (80.4%) than Hindus (70.3%) and Muslims (56.3%). The exclusive use of hygienic methods was lowest among the ST women (62.9%) and highest among the general (Others) category women (74.3%). As wealth status increased, there was a corresponding increase in the exclusive use of hygienic methods. While the proportion of women using hygienic methods was 86.0% in the wealthiest household, the proportion of such women in the poorest household was merely 48.0%.

**Table 2. Percentage distribution of urban women aged 15–24 years by background characteristics, NFHS-5 (2019–21), India.**

| Background characteristics | N (54,561) | % |
|---|---|---|
| **Respondent's current age (in years)** | | |
| 15–19 | 26,929 | 49.36 |
| 20–24 | 27,632 | 50.64 |
| **Age at menarche (in years)** | | |
| Do not remember | 369 | 0.68 |
| ≤12 | 10,871 | 19.92 |
| 13–15 | 41,200 | 75.51 |
| ≥16 | 2,121 | 3.89 |
| **Age at marriage (in years)** | | |
| Not married | 40,359 | 73.97 |
| <18 years | 3807 | 6.98 |
| ≥18 years | 10379 | 19.02 |
| Do not know | 16 | 0.03 |
| **Respondent's highest level of education** | | |
| No education | 1,826 | 3.35 |
| Primary | 2,206 | 4.04 |
| Secondary | 35,437 | 64.95 |
| Higher | 15,092 | 27.66 |
| **Religion** | | |
| Hindu | 38,368 | 70.32 |
| Muslim | 10,564 | 19.36 |
| Christian | 3,573 | 6.55 |
| Others | 2,056 | 3.77 |
| **Social Group** | | |
| Do not know | 3,176 | 5.82 |
| Scheduled Caste | 10,492 | 19.23 |
| Scheduled Tribe | 5,794 | 10.62 |
| Other Backward Classes | 22,270 | 40.82 |
| Others | 12,829 | 23.51 |
| **Household wealth status** | | |
| Poorest | 12,857 | 23.56 |
| Poorer | 12,277 | 22.5 |
| Middle | 10,985 | 20.13 |
| Richer | 10,006 | 18.34 |
| Richest | 8,436 | 15.46 |
| **Region of residence** | | |
| North | 12,893 | 23.63 |
| Central | 11,548 | 21.17 |
| East | 6,612 | 12.12 |
| West | 6,990 | 12.81 |
| Southern | 10,724 | 19.66 |
| North-east | 5,794 | 10.62 |
| **Type of home** | | |
| Marital | 12,040 | 22.07 |
| Natal | 41,870 | 76.74 |
| Other | 240 | 0.44 |

*(Continued)*

**Table 2.** (Continued)

| Background characteristics | N (54,561) | % |
|---|---|---|
| Head of the household | 411 | 0.75 |
| **Exposure to mass media** | | |
| No exposure | 4,499 | 8.25 |
| Low exposure | 23,852 | 43.72 |
| Medium exposure | 22,224 | 40.73 |
| High exposure | 3,986 | 7.31 |
| **Discussed menstrual hygiene with health worker** | | |
| No | 53,922 | 98.83 |
| Yes | 639 | 1.17 |
| **Working status of respondent** | | |
| Question not asked | 46,390 | 85.02 |
| Not working | 7,148 | 13.1 |
| Working | 1,023 | 1.87 |
| **Owns a bank account** | | |
| Question not asked | 46,390 | 85.02 |
| No | 2,107 | 3.86 |
| Yes | 6,064 | 11.11 |
| **Owns a mobile phone** | | |
| Question not asked | 46,390 | 85.02 |
| No | 3,272 | 6 |
| Yes | 4,899 | 8.98 |

Note: N = No. of sample size.

Exclusive use of hygienic methods was most prevalent among women who reported medium exposure to mass media (75.2%). It was the lowest among those who reported no exposure to mass media (44.8%). The use of exclusive hygiene methods was slightly higher among those women who met health workers and discussed menstrual hygiene with them in the three months preceding the survey than those who did not meet any healthcare worker (71.9% vs 68.1%.). Women who owned bank accounts and mobile phones showed higher use of hygienic methods (70.0% and 73.1%, respectively) than those who did not. The prevalence was higher in the southern (78.0%) and northern regions (78.0%) of India as compared to the central and north-eastern regions (50.9% and 56.8%, respectively). The results also show significant diversity in the exclusive use of hygienic methods at the regional, state, and district levels (described in the next section).

## Spatial pattern of exclusive use of hygienic methods

Analysis at the regional level provides only a broad idea regarding spatial variation in the exclusive use of hygienic methods and masks spatial heterogeneity at micro levels. Therefore, we mapped the exclusive use of hygienic methods at the state and district levels. Fig 3A presents the state-wise exclusive use of hygienic methods among young women in urban India. It is observed that the exclusive use was highest in Mizoram (92.6%), followed by Tamil Nadu (90.2%). On the other hand, it was lowest in Manipur (43.6%), Chhattisgarh (46.8%), Bihar (47.8%), and Uttar Pradesh (48.0%).

The state average obscures variation between individual districts within a state. Therefore, we analysed the patterns at the district level. Fig 3B depicts spatial patterns of exclusive use of

**Table 3. Percentage of urban women aged 15–24 who reported exclusive use of hygienic methods for menstrual bloodstain prevention, by selected background characteristics, NFHS-5 (2019–21), India.**

| Background characteristics | Percent of Women Using Hygienic methods (Weighted Percentage) N = 54,561 | P value | 95% CI | |
|---|---|---|---|---|
| | | | Lower | Upper |
| **Respondent's current age (in years)** | (16.70) | 0.004 | | |
| 15–19 | 68.93 | | 67.75 | 70.09 |
| 20–24 | 67.30 | | 66.21 | 68.37 |
| **Age at menarche (in years)** | (174.18) | <0.001 | | |
| ≤12 | 69.17 | | 67.57 | 70.72 |
| 13–15 | 67.95 | | 66.86 | 69.03 |
| ≥16 | 70.39 | | 67.26 | 73.34 |
| **Age at marriage (in years)** | (731.06) | <0.001 | | |
| Not marriage | 70.98 | | 69.94 | 72.00 |
| <18 year | 52.36 | | 49.80 | 54.90 |
| ≥18 year | 63.93 | | 62.41 | 65.43 |
| **Respondent's highest level of education** | (2753.57) | <0.001 | | |
| No education | 33.42 | | 29.78 | 37.27 |
| Primary | 39.5 | | 36.21 | 42.88 |
| Secondary | 66.82 | | 65.71 | 67.91 |
| Higher | 78.98 | | 77.87 | 80.04 |
| **Religion** | (1046.95) | <0.001 | | |
| Hindu | 70.32 | | 69.29 | 71.32 |
| Muslim | 56.31 | | 54.05 | 58.54 |
| Christian | 80.37 | | 77.08 | 83.29 |
| Others | 83.79 | | 80.86 | 86.34 |
| **Social groups** | (388.24) | <0.001 | | |
| Scheduled Caste | 65.64 | | 63.86 | 67.37 |
| Scheduled Tribe | 62.79 | | 59.44 | 66.02 |
| Other Backward Classes | 65.70 | | 64.35 | 67.02 |
| Others | 74.33 | | 72.60 | 75.99 |
| **Household wealth status** | (4080.84) | <0.001 | | |
| Poorest | 48.04 | | 46.11 | 49.98 |
| Poorer | 64.21 | | 62.58 | 65.80 |
| Middle | 70.84 | | 69.32 | 72.32 |
| Richer | 77.44 | | 76.00 | 78.82 |
| Richest | 86.02 | | 84.76 | 87.18 |
| **Region of residence** | (2976.82) | <0.001 | | |
| North | 77.97 | | 76.67 | 79.23 |
| Central | 50.9 | | 49.00 | 52.80 |
| East | 62.71 | | 59.93 | 65.42 |
| West | 74.06 | | 71.63 | 76.36 |
| South | 77.90 | | 76.28 | 79.44 |
| North-east | 56.79 | | 53.17 | 60.34 |
| **Type of home** | (389.31) | <0.001 | | |
| Marital | 60.98 | | 59.45 | 62.49 |
| Natal | 70.24 | | 69.19 | 71.27 |
| Other | 77.89 | | 68.29 | 85.21 |
| Head of the household | 68.46 | | 60.6 | 75.40 |
| **Exposure to mass media** | (1820.67) | <0.001 | | |

*(Continued)*

**Table 3.** (Continued)

| Background characteristics | Percent of Women Using Hygienic methods (Weighted Percentage) N = 54,561 | P value | 95% CI | |
|---|---|---|---|---|
| | | | Lower | Upper |
| No exposure | 44.79 | | 42.06 | 47.55 |
| Low exposure | 64.35 | | 63.04 | 65.63 |
| Medium exposure | 75.15 | | 74.07 | 76.21 |
| High exposure | 74.82 | | 72.32 | 77.17 |
| **Discussed menstrual hygiene with health worker** | (4.55) | 0.182 | | |
| No | 68.05 | | 67.05 | 69.02 |
| Yes | 71.85 | | 66.19 | 76.89 |
| **Working status of the respondent** | (2.85) | 0.620 | | |
| Not working | 67.43 | | 65.31 | 69.48 |
| Working | 66.62 | | 61.62 | 71.27 |
| **Owns a bank account** | (72.4)2 | <0.001 | | |
| No | 60.06 | | 56.60 | 63.43 |
| Yes | 69.92 | | 67.73 | 72.01 |
| **Owns a mobile phone** | (179.59) | <0.001 | | |
| No | 59.03 | | 56.04 | 61.96 |
| Yes | 73.08 | | 70.92 | 75.12 |

Note: The values in parenthesis are Chi-squared statistics. CI = confidence interval, N = sample size

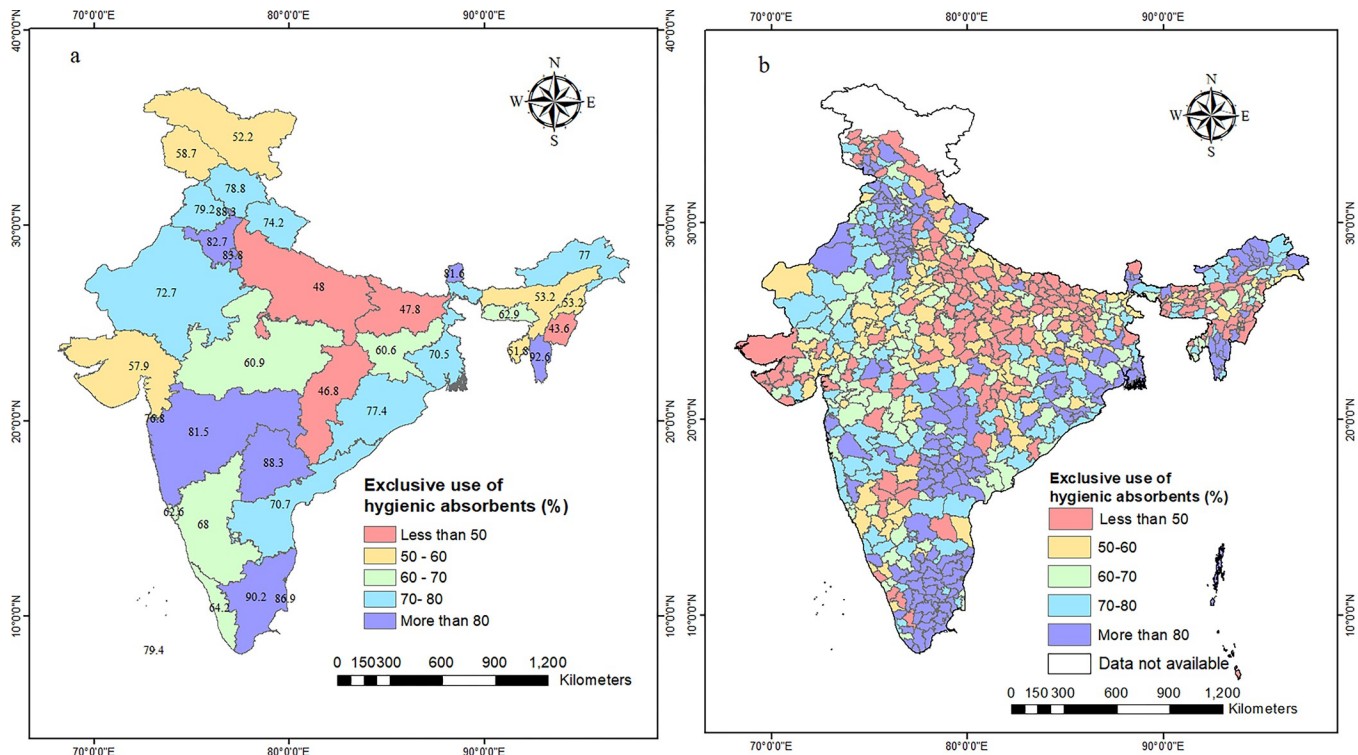

**Fig 3. Distribution of exclusive use of hygienic methods during menstruation among urban women aged 15–24 years in India, NFHS-5, 2019–21.** (a) State-wise distribution of exclusive use of hygienic methods during menstruation among urban women aged 15–24 years in India. (b) District-wise distribution of exclusive use of hygienic methods during menstruation among urban women aged 15–24 years in India. Source: authors' own creation.

hygienic methods among young urban women across all 707 districts of India. The spatial pattern of exclusive use of hygienic methods is considerably more varied than the state-level geographical pattern. The exclusive use of hygienic methods during menstruation varied among young urban women in different districts of India, with lower use reported in Kaimur district of Bihar (less than 15%) and higher use reported in Siddipet district of Telangana (more than 90%). The examination of within-state district-level patterns has revealed considerable between-district variation in the exclusive employment of hygienic methods across numerous states. For instance, in Chhattisgarh, the proportion of women reporting exclusive use of hygienic methods ranged from 26.0% in Mahasamund to 76.6% in Narayanpur. In Bihar, it varied from 11% in Kaimur to 78.8% in Madhepura, with less than 30% of exclusive use in five out of 38 districts in Bihar. Even in states with high exclusive use, such as Tamil Nadu, there were multiple districts where the prevalence was below 70%. In Karnataka, the exclusive usage ranged from 27.3% in Gadag and 33.3% in Bagalkot to 93.7% in Dakshina Kannada. In a similar vein, in Kerala, where the state average was 64.2%, the exclusive usage ranged from 35.9% in Idukki to 95.3% in Pathanamthitta.

In almost a quarter of all districts (175 out of 707), the exclusive use of hygienic methods was less than 50%. Several pockets of low exclusive use of hygienic methods were identified. The first pocket was spread over large swathes of central Indian states of Uttar Pradesh, Madhya Pradesh, Bihar, and Chhattisgarh. The second pocket was located in northeast India, comprising the districts of Assam, Manipur, Nagaland, and Meghalaya. Other pockets of low exclusive use were in western Gujarat, northern Kerala, northern Karnataka, and south-eastern Rajasthan. Within these pockets of low exclusive use, there were several districts with ultra-low exclusive use (less than 30%)—for example, Hardoi (18%), Ballia (21.3%), Lakhimpur (23.5%) districts of Uttar Pradesh and Kaimur (11%) and Gopalganj (23.8%) in Bihar, and Khandwa (17%) in Madhya Pradesh.

A little over one-fourth of all districts in the country (187 out of 707) had a prevalence of over 80%. Of these, about 10% (70 districts) had a prevalence of over 90%. There were three main pockets of high exclusive use of hygienic methods in the country, two in south India and one in northwest India. The southern pockets were spread over the district of Tamil Nadu and Telangana. The north-western pocket included almost all the districts of Delhi, Haryana and Punjab, and some southern Himachal Pradesh and northern Rajasthan districts. A small pocket of high exclusive use was also located in northern Odisha.

## Results of multivariable logistic regression

Table 3 presents the results of a Chi-squared test that investigated the relationship between exclusive use of hygienic methods and various independent variables. The findings indicated that variables such as the "working status of the respondents" and "discussion on menstrual hygiene with healthcare workers" did not demonstrate statistical significance and were, therefore, excluded from the multivariable binary logistic regression model. All other variables that were statistically significant ($p \leq 0.05$) in the Chi-squared test were included in the model. Additionally, we calculated the Variance Inflation Factors (VIFs) to detect multicollinearity among the independent variables incorporated in the regression model. The mean VIF in our study was 2.8, which falls within the acceptable range of less than 10; therefore, all variables were retained in the final model [60] (The VIF values are available on request).

Table 4 presents adjusted odds ratios obtained from the multivariable logistic regression model. The results revealed that the odds of exclusive use of hygienic methods among those women who attained menarche at 16 or later were 22% higher (AOR: 1.22, 95% CI: 1.10–1.37) than those who attained it at 12 or earlier. Further, the results show that the odds of using

**Table 4.** Logistic regression analysis showing the factors associated with the exclusive use of hygienic methods among urban women aged 15–24 years in India NFHS-5 (2019–21).

| Background characteristics | Adjusted odds ratio | P value | 95% CI | |
|---|---|---|---|---|
| | | | Lower | Upper |
| **Respondent's current age (in years)** | | | | |
| 15–19 (Reference) | | | | |
| 20–24 | 0.82 | <0.001 | 0.78 | 0.86 |
| **Age at menarche (in years)** | | | | |
| ≤12 (Reference) | | | | |
| 13–15 | 1.07 | 0.012 | 1.01 | 1.12 |
| ≥16 | 1.22 | <0.001 | 1.10 | 1.37 |
| **Age at marriage (in years)** | | | | |
| Not marriage (Reference) | | | | |
| <18 year | 0.70 | <0.001 | 0.63 | 0.78 |
| ≥18 year | 0.85 | <0.001 | 0.77 | 0.93 |
| **Respondent's highest level of education** | | | | |
| No education (Reference) | | | | |
| Primary | 1.28 | <0.001 | 1.12 | 1.47 |
| Secondary | 2.18 | <0.001 | 1.96 | 2.43 |
| Higher | 3.02 | <0.001 | 2.68 | 3.40 |
| **Religion** | | | | |
| Hindu (Reference) | | | | |
| Muslim | 0.61 | <0.001 | 0.58 | 0.64 |
| Christian | 1.34 | <0.001 | 1.20 | 1.49 |
| Others | 1.22 | 0.001 | 1.09 | 1.37 |
| **Social groups** | | | | |
| Scheduled Caste (Reference) | | | | |
| Scheduled Tribe | 1.04 | 0.412 | 0.95 | 1.13 |
| Other Backward Classes | 0.91 | 0.001 | 0.86 | 0.96 |
| Others | 1.20 | <0.001 | 1.12 | 1.27 |
| **Household wealth status** | | | | |
| Poorest (Reference) | | | | |
| Poorer | 1.65 | <0.001 | 1.57 | 1.75 |
| Middle | 2.14 | <0.001 | 2.02 | 2.27 |
| Richer | 2.83 | <0.001 | 2.65 | 3.03 |
| Richest | 4.38 | <0.001 | 4.05 | 4.74 |
| **Region of residence** | | | | |
| Central (Reference) | | | | |
| North | 2.41 | <0.001 | 2.27 | 2.56 |
| East | 2.09 | <0.001 | 1.95 | 2.24 |
| West | 1.68 | <0.001 | 1.57 | 1.80 |
| South | 3.15 | <0.001 | 2.96 | 3.35 |
| North-east | 1.84 | <0.001 | 1.69 | 2.01 |
| **Type of home** | | <0.001 | | |
| Marital (Reference) | | | | |
| Natal | 0.93 | 0.103 | 0.85 | 1.02 |
| Other | 0.92 | 0.585 | 0.68 | 1.25 |
| Head of the household | 1.35 | 0.010 | 1.07 | 1.69 |
| **Exposure to mass media** | | | | |

*(Continued)*

**Table 4.** (Continued)

| Background characteristics | Adjusted odds ratio | P value | 95% CI | |
|---|---|---|---|---|
| | | | Lower | Upper |
| No exposure (Reference) | | | | |
| Low exposure | 1.29 | <0.001 | 1.20 | 1.38 |
| Medium exposure | 1.34 | <0.001 | 1.24 | 1.44 |
| High exposure | 1.24 | <0.001 | 1.12 | 1.37 |
| **Owns a bank account** | | <0.001 | | |
| No (Reference) | | | | |
| Yes | 1.05 | 0.368 | 0.94 | 1.18 |
| **Owns a mobile phone** | | | | |
| No (Reference) | | | | |
| Yes | 1.25 | <0.001 | 1.12 | 1.39 |

Note: CI = Confidence interval

hygienic methods among those who women who were married off before the legal age of 18 years were about 30% lower (AOR: 0.70, 95% CI: 0.63–0.78) than unmarried women.

The odds of exclusive use of hygienic methods varied considerably by the level of respondents' education. The odds of exclusive use among women with secondary education and higher education were two (AOR: 2.18, 95% CI: 1.96–2.43) and three times (AOR: 3.02, 95% CI: 2.68–3.40) higher than those with no education. The odds among Muslim women were 40% lower (AOR: 0.61, 95% CI: 0.58–0.64 than their Hindu counterparts. The odds of using hygienic methods were 20% higher among others (AOR: 1.20, 95% CI: 1.12–1.27) than they are among SCs.

The wealth status exhibits a positive effect on the exclusive use of hygienic methods among urban women. The odds of exclusive use among women from the richest wealth quintile were nearly four times (AOR: 4.38, 95% CI: 4.05–4.74) higher than those from the poorest wealth quintile. The odds also varied significantly across the regions of India. For example, women from the south (AOR: 3.15, 95% CI: 2.96–3.35) and the north regions (AOR:2.41, 95% CI: 2.27–2.56) had higher odds of exclusive use of hygienic methods than women from the central region.

The odds of exclusive use among women who were the head of the households were 35% higher than those who resided in their marital homes (AOR: 1.35, 95% CI: 2.27–2.56). The odds of using hygienic methods among urban women living in their marital and natal homes were not significantly different. However, the odds for women with low, medium, and high mass media exposure were around 1.3 times higher than those without mass media exposure.

## Discussion

This paper aimed to identify the differentials and correlates of the exclusive use of hygienic methods during menstruation among young women in urban India. Two out of every three young women in urban India reported exclusive use of hygienic methods during menstruation. However, there was significant geographic heterogeneity in the exclusive use of hygienic methods at both state and district levels. The findings of this study revealed the existence of disadvantaged subgroups among urban women who require targeted interventions from policymakers through the implementation of context-specific programs and policies. The results highlighted that women's education, mass media exposure, household wealth, and religion were positively linked to the exclusive use of hygienic methods during menstruation among

urban women. Additionally, the analysis uncovered statistically significant associations between the exclusive use of hygienic methods and biodemographic factors such as age, age at menarche, and age at marriage.

In the current study, the exclusive use hygienic methods during menstruation were higher among younger women (aged 15–19) than their older counterparts (aged 20–24). One possible explanation is the proliferation of programs that offer teenage girls subsidized or free access to hygienic methods while they are still in school [61–63]. The prevalence of using hygienic methods increases with the rising age of menarche and marriage because women become increasingly aware of the various hygienic methods available to them as they age. A previous study reported that women having menarche at a younger age had insufficient knowledge of menstruation and the importance of using hygienic methods [64].

The findings of the current study suggest that urban women with higher education had a higher propensity to use hygienic methods exclusively than illiterate women. This finding is in line with existing evidence on the subject [18, 32, 41]. Women with a higher level of education are more cautious of the health risks of unhygienic menstrual practices, have more decision-making autonomy, and are often financially more independent [32, 53]. The education also provides opportunities to learn about reproductive health through exposure to mass media [11]. Consistent with prior research, Muslim women demonstrated a lower propensity to use hygienic methods than other religions, possibly due to a lack of mobility rights, limited information and awareness, and religious taboos on the disposal and storage of hygienic methods in the Muslim population [65].

The disparity in the exclusive use of hygienic methods across different economic/wealth groups has been a concern of many researchers. Several studies have documented the fact that household wealth has a positive effect on the exclusive use of hygienic methods [32, 33, 65]. The study affirms that young women in urban India exhibit similar trends, with those from the wealthiest households having four times greater odds of using hygienic methods compared to those from the poorest households. It has been evidenced that wealthier urban women have enough resources to buy hygienic methods. In contrast, poor women are generally less educated, often unemployed or stay-at-home mothers, and may struggle to afford the more expensive hygienic methods [33].

It must be noted here that one-third of the poor urban population in India resides in slums [66]. Lack of adequate housing and essential public amenities are the issues that have disproportionately affected the poor slum-dwellers [67, 68]. The inability to properly manage menstruation is a challenge for women and girls living in urban slums due to the factors mentioned above, as well as the lack of privacy, unsafe environments, vulnerability, inadequate infrastructure and services, poor water, sanitation and hygiene (WASH) facilities, and inadequate knowledge and practice of menstrual hygiene [69, 70]. Understanding the challenges women and girls face and enabling gender-sensitive policy and programme interventions that address their sanitation and hygiene needs is particularly timely in light of the poor living and sanitation conditions in slum areas and the current policy focus on WASH [70].

The present study revealed that increased mass media exposure increased the exclusive use of hygienic methods. This finding is consistent with several prior Indian and international investigations [32, 39]. The mass media could serve as reliable sources of information, enhancing women's understanding of the health benefits of using hygienic methods and raising awareness of those available at subsidized prices [33, 53].

In this study, the ownership of mobile phones was positively associated with the exclusive use of hygienic methods during menstruation. The greater availability of mobile phones has brought about several advantages and opportunities for young women today. These mobile phones can potentially improve access to healthcare and understanding of the importance of

menstrual hygiene and general wellbeing. Previous studies have also linked mobile phone ownership with digital awareness and increased use of exclusive hygienic methods [41].

State and district-level variations in the exclusive use of hygienic methods during menstruation among young urban women must be considered during policymaking and programme implementation. A possible explanation for this district-level variation could be the implementation of various subsidized and free sanitary napkin distribution initiatives. For example, the state of Tamil Nadu, where exclusive use of hygienic methods is over 80%, has provided 20 sanitary napkins free of cost to peri-urban women since 2011 under a scheme called *Pudhu Yugam* (New Era) [71]. The *'Swechha'* (Freedom) scheme of the Government of Andhra Pradesh provides free and subsidized sanitary napkins to girls in classes seven through ten. *Ruthu Prema* (Safe Periods) is a new program in Telangana that encourages using menstruation cups and provides free sanitary pads to all women via Anganwadi workers and municipal officers. Other states with high exclusive use of hygienic methods, such as Maharashtra (*Asmita*) and Punjab (*Udaan*), have also instituted free or subsidized sanitary napkin distribution programs. Moreover, in some areas of these states, schools have placed sanitary napkin vending machines in conjunction with local NGOs, dispensing locally produced napkins at a discounted rate [72].

Kerala (She Pad), Karnataka *(Suchi)*, Gujarat (*Taruni Suvidha)*, Odisha *(Khushi)*, and Uttarakhand (*Sparsh*) have also launched subsidized sanitary napkin schemes to promote the use of hygienic methods during menstruation among young girls and women [73–75]. However, it is unclear why some districts' exclusive use of hygienic methods is substantially lower than other districts in these states. Further research is needed to unearth the reasons behind such within-state spatial disparities in the exclusive use of hygienic methods during menstruation among young urban women in these states.

Rajasthan (*Udaan*), Bihar (*Kanya Utthan Yojana*), Uttar Pradesh (*Kishori Suraksha Yojana*), Madhya Pradesh (*Udita*), Chhattisgarh (*Suchita*) Assam, Tripura (*Kishori Suchita Abhiyan*) and Manipur (*My Pad My Right*), also have implemented schemes to promote sanitary napkin use during menstruation [76–79]. In spite of this, a small percentage of young urban women in these states exclusively use hygienic methods during menstruation. It is most likely because these initiatives and programs run into several challenges, including supply and procurement problems, low-quality hygienic methods, a lack of knowledge and comprehension of the programs, financial irregularities, greater costs, and widespread corruption [80–82]. These challenges, however, are mainly organizational and can be dealt with by some structural changes.

The Government of India has also made several efforts to promote menstrual hygiene management among urban women via National Health Mission sponsored programmes. For example, the Central Government has set up a network of over 8500 subsidized pharmacies known as *Jan Aushadhi Kendras* (JAK) in all 640 districts [83]. However, this is insufficient for a population of 1.4 billion. In 2020, the government launched a brand of sanitary napkins known as *Suvidha* (Convenience), composed entirely of biodegradable materials and sold at a discounted price at these pharmacies [13, 84]. These programmes, however, have been hampered by procurement and supply issues, high costs, and a lack of an effective distribution mechanism [85]. These factors could be some of the reasons why the use of sanitary methods among young urban women in the vast majority of the country's districts continues to remain lower than expected [86].

This study sheds light on the exclusive use of hygienic methods during menstruation among young urban women in India, but there are several limitations that need to be considered. Firstly, as this is a cross-sectional study, the causal relationship between predictor and outcome variables cannot be established. Moreover, the analysis solely focuses on demand-

side factors, neglecting the potential impact of supply-side variables on the use of menstrual products. The absence of supply-side variables in the NFHS dataset means that factors such as the availability and pricing of hygienic methods at urban pharmacies and provision stores, as well as the provision of sanitary napkins in schools, could not be incorporated into the analysis, potentially influencing the study's results. Additionally, the NFHS does not provide any information on the unmet need of menstrual materials or access to menstrual products in India. Further investigation is necessary to uncover the unmet needs of menstrual materials. Also, the survey does not provide any information on the materials used in locally prepared napkins. Furthermore, the NFHS data does not provide any information on whether women are properly sanitizing or washing their reusable menstrual hygiene materials, making it difficult to determine whether they are using such materials in a hygienic manner. The survey also does not cover critical factors such as social taboos, cultural norms and traditions, and disposability, which could potentially impact the use of hygienic methods [87]. Therefore, further research is needed to examine these factors and establish how they affect the use of hygienic methods. Furthermore, due to data constraints, this study does not include slum and non-slum variable in the analysis to determine intra-urban differences. Finally, the reasons behind the low exclusive use of hygienic methods in certain districts of some states with high averages remain unclear, and more investigation is needed to uncover these reasons. Overall, while this study provides valuable insights, these limitations highlight the need for further research to fully understand the factors influencing the use of menstrual hygiene materials among women in India.

## Conclusion

This study highlights the existence of significant socioeconomic, biodemographic, and geographic disparities in the use of hygienic menstrual methods among young women in urban India. To address these disparities, there is a need for advocacy campaigns, mass media exposure, educational outreach, and subsidized or free sanitary napkins for urban women, particularly those from underserved castes, tribes, and religions. Although there have been recent initiatives by central and state governments to improve access to hygienic menstrual products, many of these programs remain in the pilot phase or are limited to certain areas. Therefore, it is crucial to expand these initiatives to reach as many underserved individuals as possible. Additionally, governments must acknowledge the micro-level (district) disparities in the use of hygienic menstrual methods among urban women and focus on specific geographies highlighted in this study.

## Acknowledgments

The authors are grateful to the Demographic and Health Surveys (DHS) for providing the dataset for this study. Dr. Aditya Singh acknowledges the support provided by Banaras Hindu University's Institute of Eminence (IoE) Seed Grant No. R/Dev/D/IoE/Equipment/Seed Grant-II/2022-23/48726. This paper is a part of Mahashweta Chakrabarty's PhD research work.

## Author Contributions

**Conceptualization:** Aditya Singh.

**Data curation:** Mahashweta Chakrabarty.

**Formal analysis:** Aditya Singh, Mahashweta Chakrabarty.

**Methodology:** Aditya Singh, Mahashweta Chakrabarty.

**Resources:** Mahashweta Chakrabarty.

**Supervision:** Aditya Singh.

**Validation:** Aditya Singh, Rakesh Chandra, Sourav Chowdhury.

**Visualization:** Aditya Singh, Mahashweta Chakrabarty.

**Writing – original draft:** Aditya Singh, Mahashweta Chakrabarty.

**Writing – review & editing:** Aditya Singh, Mahashweta Chakrabarty, Rakesh Chandra, Sourav Chowdhury, Shivani Singh.

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
