## [Decision Letter · Decision Letter 0]

26 Sep 2022

PGPH-D-22-01066

Intra-urban differentials in the exclusive use of hygienic absorbents among menstruating women in India: Evidence from NFHS-5

Dear Dr. Singh,

Thank you for submitting your manuscript to PLOS Global Public Health. After careful consideration, we feel that it has merit but does not fully meet PLOS Global Public Health’s publication criteria as it currently stands. Therefore, we invite you to submit a revised version of the manuscript that addresses the points raised during the review process.

The manuscript has been evaluated by three reviewers, and their comments are available below.

The reviewers have raised concerns regarding the rationale, reporting and methodology of this study. 

Could you please revise the manuscript to carefully address the concerns raised?

We look forward to receiving your revised manuscript.

Kind regards,

Johannes Stortz

Staff Editor

Journal Requirements:

2. Some material included in your submission may be copyrighted. According to PLOS’s copyright policy, authors who use figures or other material (e.g., graphics, clipart, maps) from another author or copyright holder must demonstrate or obtain permission to publish this material under the Creative Commons Attribution 4.0 International (CC BY 4.0) License used by PLOS journals. Please closely review the details of PLOS’s copyright requirements here: PLOS Licenses and Copyright. If you need to request permissions from a copyright holder, you may use PLOS's Copyright Content Permission form.

Potential Copyright Issues:

Figure 3: please (a) provide a direct link to the base layer of the map (i.e., the country or region border shape) and ensure this is also included in the figure legend; and (b) provide a link to the terms of use / license information for the base layer image or shapefile. We cannot publish proprietary or copyrighted maps (e.g. Google Maps, Mapquest) and the terms of use for your map base layer must be compatible with our CC-BY 4.0 license. 

Additional Editor Comments (if provided):

Reviewers' comments:

Reviewer's Responses to Questions

**Comments to the Author**

1. Does this manuscript meet PLOS Global Public Health’s publication criteria? Is the manuscript technically sound, and do the data support the conclusions? The manuscript must describe methodologically and ethically rigorous research with conclusions that are appropriately drawn based on the data presented.

Reviewer #1: Yes

Reviewer #2: Yes

Reviewer #3: Partly

2. Has the statistical analysis been performed appropriately and rigorously?

Reviewer #1: Yes

Reviewer #2: Yes

Reviewer #3: Yes

3. Have the authors made all data underlying the findings in their manuscript fully available (please refer to the Data Availability Statement at the start of the manuscript PDF file)?

Reviewer #1: Yes

Reviewer #2: Yes

Reviewer #3: Yes

4. Is the manuscript presented in an intelligible fashion and written in standard English?

Reviewer #1: Yes

Reviewer #2: Yes

Reviewer #3: Yes

5. Review Comments to the Author

Reviewer #1: The paper investigated the pattern of hygienic absorbents among menstruating women in urban India. The analysis is done and presented rigorously. The discussion section is excellent with all govt programs on the topic. However, if they can include slum and non slum variable in their analysis to see intra urban difference, the quality of the paper will be improved.

Reviewer #2: The study is based on the latest data set of NFHS-5 and highlights few on the novel points in the mensural hygiene research domain. The statistical methods used are popular and unambiguous. The manuscript however contains a few factual errors and omissions in the text, however amendable. I would recommend it for publication with the following corrections.

The study missed citing some of previous studies on this subject from India. I suggest authors to review those studies and nuance their discussion section.

Goli, S., Sharif, N., Paul, S., & Salve, P. S. (2020). Geographical disparity and socio-demographic correlates of menstrual absorbent use in India: A cross-sectional study of girls aged 15–24 years. Children and Youth Services Review, 117, 105283.

Malhotra, A., Goli, S., Coates, S., & Mosquera-Vasquez, M. (2016). Factors associated with knowledge, attitudes, and hygiene practices during menstruation among adolescent girls in Uttar Pradesh. Waterlines, 35. https://doi.org/10.3362/1756-3488.2016.021.

Also, add some discussion on why the question of mensural hygiene for urban setting is important. What urban processes are pushing back the young girls in accessing and using mensural absorbents. In particular, the role of housing and access to WASH facilities.

Line 31: The idea of an urban advantage in health and healthcare should be elaborated and a few references on it might be helpful.

Line 74: The authors should justify the reason behind limiting the research to 15-24 age group.

Line 96: The authors have probably calculated Variance Inflation Factors (VIF) but refer it as Variance Influence Factors (VIF). If it’s just a typographical error it should be corrected, otherwise, they should elaborate the idea of Variance Influence Factors (VIF).

Line 105: The sentence in the line seems to be incomplete and has a misplaced period sign. This should be rewritten.

Line 124: “Women using both hygienic and unhygienic absorbents” in different spells are coded as "users of unhygienic absorbents". Is there any way possible to segregate these two categories if yes it may be disaggregated?

Line 126: The authors should include some variables form NFHS data that measure the women access/barriers in getting hygienic absorbents for examples “distance travelled to buy sanitary napkins”.

Line 183: The sentence “At the district level, urban adolescent women have more geographical diversity.” looks incomplete and does not add any information it must be corrected.

Line 284: The study shows perfect 100% use of hygienic absorbents by the women in the Siddipet district of Telangana state. This seems to an outlier in the study and must be explained in a couple of subsequent lines in the text.

Figure 3.b: Some part of the map is kept blank if its due to unavailability of the data or for any other reasons the details should be given in a note.

Table 3 and 4: In both of these table the p-values has been specified up to two decimal points, these should be corrected and authors should specify these values up to three decimal points.

Reviewer #3: This manuscript uses data from the NFHS 5 to describe associations between socio-demographic characteristics and the exclusive use of commercial menstrual products (termed 'hygienic absorbents' by the authors) compared to the non-exclusive use of these products and use of other improvised absorbents such as cloth.

I have five key concerns, noted below.

(1) Multiple, very similar analyses and papers have been published on this same research question using the NFHS-4 data. The authors have not addressed these past studies in the background, indeed they claim that only small scale studies have been done.

A very quick google-scholar search will show multiple papers on the NFHS 4 data

Here's a few (I expect there are more):

Avijit Roy, Pintu Paul, Jay Saha, Bikash Barman, Nanigopal Kapasia & Pradip Chouhan (2021) Prevalence and correlates of menstrual hygiene practices among young currently married women aged 15–24 years: an analysis from a nationally representative survey of India, The European Journal of Contraception & Reproductive Health Care, 26:1, 1-10, DOI: 10.1080/13625187.2020.1810227

Kathuria, B., & TP, S. R. (2022). Factors Explaining Regional Variations in Menstrual Hygiene Practices among Young Women in India: Evidence from NFHS-4.

Maharana, B. (2022). What explains the rural-urban gap in the use of hygienic methods of menstrual protection among Youth in the East Indian state of Bihar?. Indian Journal of Community Medicine, 47(2), 182.

Vishwakarma, D., Puri, P., & Sharma, S. K. (2021). Interlinking menstrual hygiene with Women's empowerment and reproductive tract infections: Evidence from India. Clinical Epidemiology and Global Health, 10, 100668.

Ram, U., Pradhan, M. R., Patel, S., & Ram, F. (2020). Factors associated with disposable menstrual absorbent use among young women in India. International Perspectives on Sexual and Reproductive Health, 46, 223-234.

This study is the first I've seen on the NFHS-5 data, which is more recent that the NFHS-4. However, the authors then need to justify what THIS study would add to our understanding of menstrual health and hygiene, and how it relates to the myriad of past studies on this exact same research question using the previous survey.

I am very concerned that the authors haven't contextualized this study in this past work.

(2) There is (necessary) increasing emphasis in menstrual health research on the importance of product choice. UNICEF's 2019 report on menstrual health and hygiene, and the supplement on menstrual materials outlines this quite well. Indeed, authors should note that the JMP MICS reports have reported national level information on menstrual product use - their indicator includes cloth as absorbent, recognizing this is the preference of some women.

This isn't to say the authors should do the same in the analysis, but that any manuscript needs to engage with this thinking and highlight WHY we would be interested only in commercial product use, in contrast to women having enough menstrual materials, or preferred materials.

Along these lines, I find the position of the paper that the goal should be all women using commercial products, rather than that women have access to enough information to make informed decisions and access to sufficient menstrual materials.

This is a point also outlined in the recent Global Menstrual Collective definition of Menstrual Health, and the WHO statement of menstrual health and rights.

(3) I know this is likely due to what was asked in the survey, but there doesn't seem to be any differentiation between commercially produced reusable menstrual pads, and disposables. With a view to the potential environmental and cost sustainability of reusable materials, it is unclear why these aren't differentiated.

It isn't clear what 'locally prepared napkins' refers to - are these reusable or does this refer to disposables manufactured locally? There are also reusables manufactured commercially at scale.

(4) In such a large data set any differences are going to be statistically significant. Given that the outcome is a non-rare event, the odds ratios are difficult to interpret in terms of how MEANINGFUL these differences between socio-demographic characteristics are. I am not convinced at the value of these differences for practice.

'Conceptual framework' for only socio-demographic predictors seems overstated.

In a few places authors interpret odds ratios as meaning some women are 'more likely' to use hygienic absorbents - this is not correct interpretation of an odds ratio. No risk ratios are presented.

In the discussion the authors do a great job of describing many different initiatives across India to support menstrual product access. Can the NSFH data not be used to evaluate whether these initiatives have resulted in increased uptake, given we know there is a baseline from survey 4, and now data from survey 5?

Again, it is highly problematic that the article doesn't engage with the past work on this.

(5) I am concerned at some selective use of evidence in the Background section. The authors state that use of 'unhygienic absorbents' results in RTIs and school absenteeism.

But the relationships between menstrual hygiene practices and needs and these outcomes is highly mixed. Indeed the Austrian study cited found no effect of a pad-distribution intervention on girls' absenteeism. A series of work by Das and Torondel in Odisha has also noted varied relationships between different menstrual hygiene practices and RTI.

We do hypothesize relationships between unmet menstrual health and hygiene needs and these outcomes, but the literature and relationships between these concepts is much more complex (and more researched) than the brief background provided here.

Again, I am not sure 'unhygienic absorbents' is fairly applied to all non-commercial product use.

6. PLOS authors have the option to publish the peer review history of their article (what does this mean?). If published, this will include your full peer review and any attached files.

**Do you want your identity to be public for this peer review?** For information about this choice, including consent withdrawal, please see our Privacy Policy.

Reviewer #1: No

Reviewer #2: No

Reviewer #3: No

---

## [Decision Letter · Decision Letter 1]

13 Mar 2023

PGPH-D-22-01066R1

Intra-urban differentials in the exclusive use of hygienic methods during menstruation among young women in India

Dear Dr. Singh,

Thank you for submitting your manuscript to PLOS Global Public Health. After careful consideration, we feel that it has merit but does not fully meet PLOS Global Public Health’s publication criteria as it currently stands. Therefore, we invite you to submit a revised version of the manuscript that addresses the points raised during the review process.

The manuscript has been evaluated by three reviewers, and their comments are available below.

The concerns raised by Reviewers 1 and 2 have been addressed. However, reviewer #3 has provided additional comments regarding concerns that have not been addressed sufficiently.

Could you please revise the manuscript to carefully address the concerns raised?

We look forward to receiving your revised manuscript.

Kind regards,

Richard Ali, PhD

Staff Editor

Journal Requirements:

Additional Editor Comments (if provided):

Reviewers' comments:

Reviewer's Responses to Questions

**Comments to the Author**

1. If the authors have adequately addressed your comments raised in a previous round of review and you feel that this manuscript is now acceptable for publication, you may indicate that here to bypass the “Comments to the Author” section, enter your conflict of interest statement in the “Confidential to Editor” section, and submit your "Accept" recommendation.

Reviewer #3: (No Response)

2. Does this manuscript meet PLOS Global Public Health’s publication criteria? Is the manuscript technically sound, and do the data support the conclusions? The manuscript must describe methodologically and ethically rigorous research with conclusions that are appropriately drawn based on the data presented.

Reviewer #3: Yes

3. Has the statistical analysis been performed appropriately and rigorously?

Reviewer #3: Yes

4. Have the authors made all data underlying the findings in their manuscript fully available (please refer to the Data Availability Statement at the start of the manuscript PDF file)?

Reviewer #3: Yes

5. Is the manuscript presented in an intelligible fashion and written in standard English?

Reviewer #3: Yes

6. Review Comments to the Author

Reviewer #3: The manuscript has been strengthened through author attention to the peer reviewer comments.

I appreciate the authors expanding attention in the background to disparities among urban populations.

However I do not feel all of the comments have been sufficiently addressed.

1 - there are still places in the manuscript that report odds ratios as if they are risk ratios and state that respondents were 'more likely' to have a particular outcome. This is an important misrepresentation of statistics and the manuscript cannot be published before this is addressed through out

e.g., page 8 "Urban women who got their first period after age 16 were slightly more likely to use hygienic methods than those who got their first period earlier."

What do we mean by 'slightly' more likely. In my first comments to the authors I highlighted that any differences among the groups were likely to be statistically significant given the sample size. "more likely" appears again on page 13 and throughout other places.

2 - The authors acknowledged in their reply that the relationship between menstrual material use and urogenital infection/RTI is complex and underresearched. Authors have added some citations around this in the paper but have still included strong statements assuming that the use of cloth (as defined as 'unhygenic') causes negative outcomes.

This is present even in the revised abstract

"The use of hygienic methods during menstruation is substantially lower among urban women in India than in many other developing countries, resulting in higher reproductive morbidity."

As highlighted in my past review, and acknowledged by the authors, this statement is not sufficiently supported by current evidence and should be tempered.

3 - I understand that the authors want to use the same language as the NFHS - I still feel very uncomfortable that the authors are referring to cloth as 'unhygienic' - it evokes a judgement on women that I feel adds to the stigma around menstruation. As the authors note, cloth could be considered hygienic if cleaned appropriately. I suggest the authors consider if 'hygienic' is the language they want to use - and if so perhaps add some reflective text on this in the introduction.

4 - Authors have responded to my comment about informed choice by including some mentions of this in the discussion. I can't see attention to understanding if women have ENOUGH menstrual materials. Notably research has found that use of commercial pads doesn't necessarily imply that women have enough materials.

https://www.mdpi.com/1660-4601/17/8/2633

This study found that across 10 countries, among respondents exclusively using pads 26% still reported having unmet menstrual material needs.

The limitations section of the manuscript should highlight not only the limitations of the cross sectional design - but also be clear about what this data can and can't tell us about menstrual product access & menstrual health in India.

7. PLOS authors have the option to publish the peer review history of their article (what does this mean?). If published, this will include your full peer review and any attached files.

**Do you want your identity to be public for this peer review?** For information about this choice, including consent withdrawal, please see our Privacy Policy.

Reviewer #3: No

---

## [Editor Report · Decision Letter 2]

22 May 2023

Intra-urban differentials in the exclusive use of hygienic methods during menstruation among young women in India

PGPH-D-22-01066R2

Dear Ms. Chakrabarty,

We are pleased to inform you that your manuscript 'Intra-urban differentials in the exclusive use of hygienic methods during menstruation among young women in India' has been provisionally accepted for publication in PLOS Global Public Health.

Best regards,

Tia M. Palermo

Academic Editor